# Insights into the Substrate Uptake Mechanism of Mycobacterium Tuberculosis Ribose 5-Phosphate Isomerase and Perspectives on Drug Development

Leonardo Bartkevihi [1,2], Ícaro P. Caruso [2,3], Bruna Martins [4], José R. M. Pires [1], Danielle M. P. Oliveira [4], Cristiane Dinis Anobom [4,*] and Fabio C. L. Almeida [1,2,*]

1   Institute of Medical Biochemistry (IBqM), National Center of Nuclear Magnetic Resonance Jiri Jonas, Federal University of Rio de Janeiro, Rio de Janeiro 21941-902, Brazil
2   National Center of Nuclear Magnetic Resonance (CNRMN), Center for Structural Biology and Bioimaging (CENABIO), Federal University of Rio de Janeiro, Rio de Janeiro 21941-902, Brazil
3   Multiuser Center for Biomolecular Innovation (CMIB), Department of Physics, São Paulo State University (UNESP), São José do Rio Preto 15054-000, Brazil
4   Laboratório de Bioquímica Estrutural de Proteínas (LaBEP), Institute of Chemistry, Department of Biochemistry, Federal University of Rio de Janeiro, Rio de Janeiro 21941-617, Brazil
*   Correspondence: anobom@iq.ufrj.br (C.D.A.); falmeida@bioqmed.ufrj.br (F.C.L.A.)

**Abstract:** The active site of the dimeric ribose 5-phosphate isomerase B (RpiB) contains a solvent-exposed barrier formed by residues H12, R113, R137, and R141, which is closed upon the complexation of phosphate. The substrate ribose 5-phosphate (R5P) has to overcome the surface barrier to reach an internal cavity and then bind in the linear configuration of ribose to the interface between the two subunits. NMR and molecular dynamics simulation are suitable methods to describe the transient nature of the RpiB active site and help our understanding of the mechanism of substrate entrance. In this study, we show that the entrance of the nucleotides AMP/ADP into the internal cavity of mycobacterium tuberculosis RpiB (MtRpiB) does not involve a canonical open/close-lid conformational transition usually observed in many enzymes. Instead, a flipping mechanism in which the nucleotide phosphate interacts with the surface barrier followed by the flip of the nitrogenous base and ribose is responsible for changing the substrate/ligand orientation from a solvent-exposed to a buried state. Based on these results, we propose a substrate/inhibitor uptake mechanism that could provide a basis for rational drug design using MtRpiB, which is an essential enzyme and a good target for drug development.

**Keywords:** ribose 5-phosphate isomerase; adenosine monophosphate; substrate uptake; nuclear magnetic resonance (NMR); molecular dynamics simulations

## 1. Introduction

The development of new enzyme inhibitors and the understanding of the enzymatic mechanism often involves the structure determination of the enzyme–substrate/inhibitor complexes. Nevertheless, the depiction of intermediate transient states that lead to the complex formation is as just important and frequently overlooked. These studies are a challenge due to the small percentage of these states compared to the bound state. NMR and molecular dynamics simulations can be combined to indirectly analyze these states in enzymatic systems [1–3]. The conformational dynamics of the protein target and transient states in the pathway to binding are closely related to the biological functions, and this information must be applied to drug design and development. Additionally, the prior knowledge of the mechanism of interaction, observing all the details about the active site, cavities, and interaction loops that may appear transiently or permanently on the protein's surface during the interaction are essential [3,4].

Ribose 5-phosphate isomerase (Rpi; EC 5.3.1.6) is the first enzyme of the non-oxidative branch of the pentose phosphate pathway. It is responsible for the interconversion between ribulose 5-phosphate and ribose 5-phosphate, essential to synthesizing nucleotides, cofactors, and some amino acids [5]. Knockout and knockdown experiments have reported Rpi essentiality in organisms, such as *Escherichia coli*, trypanosomatids, and *Mycobacterium tuberculosis* [6–8]. There are two families of ribose 5-phosphate isomerase, RpiA, and RpiB. RpiA is present in most eukaryotes, while RpiB is observed almost exclusively in prokaryotes, some basal eukaryotes, and fungi. Both are structurally unrelated but catalyze the same enzymatic reaction [9]. The structural differences between human RpiA and *M. tuberculosis* RpiB (MtRpiB), only an 11.43% sequence identity, and its crucial role in cellular metabolism make this enzyme a potential drug target [10,11].

RpiBs have a Rossmann fold composed of α-helices intercalated with β-strands. These β-strands interact to form a central β-sheet protected from exposure to the solvent by the presence of α-helices on both sides. The MtRpiB is dimeric, although many RpiBs are also found as tetramers [7]. Dimerization is essential for the catalytic activity because the active site is located at the dimer interface, sharing the catalytic residues from both monomers. As several high-resolution structures are available in the literature and the catalytic mechanism is known, it is possible to identify the residues of the active site that are directly involved in catalysis or just integrate the cavity of the site [6,7,11]. The substrate ribose 5-phosphate (R5P) has to overcome the surface barrier to reach an internal cavity and then bind in the linear configuration of ribose to the interface between the two subunits. The residues of the surface barrier (H12, R113, R137, and R141) are conserved and are complexed by phosphate. The histidines H102 and H138 are involved in opening the R5P furanose ring. E75 transfers protons between C1 and C2 of the substrates, while S71 transfers the protons linked to O1 and O2. Other conserved atoms that comprise the active site are D11, Y46, and N103 [11]. Since the substrate is hidden in the dimer interface, the active site is not accessible. The transient states that drive the substrate to the dimer interface are unknown.

The identification of substrates and ligands for RpiBs is described in the literature. Some studies showed that ribulose 5-phosphate is the preferential Trypanosoma brucei RpiB substrate, with a lower Km than ribose 5-phosphate [12]. Additionally, Escherichia coli RpiB (EcRpiB) can catalyze the conversion of D-allose-6-phosphate (All6P) to D-allulose-6-phosphate (Allu6P), both six-carbon sugars, and MtRpiB is inhibited by D-allose-6-phosphate [13]. The EcRpiB structure in complex with these sugars during the formation of the cis-1,2-enediol-(ate) intermediate was essential to developing All6P and Allu6P derivatives with MtRpiB inhibitory activity [14]. Previously, the structure of high-energy intermediates during the ribose 5-phosphate catalysis by RpiBs was applied to the design of compounds that have inhibitory activity against MtRpiB, such as 4-phospho-D-erythronohydroxamic acid (4PEH) and 4-phospho-D-erythronate (4PEA) [10]. Thus, the identification of new ligands, the investigation of the structural features, and the transient state, as the information regarding the catalytic mechanism, are crucial to developing inhibitors.

In this study, we describe the interaction of MtRpiB with AMP and derivatives. The results show that phosphate is essential for the interaction with the ligands tested. Additionally, the affinity decreases by adding a phosphate to ADP, indicating that ATP mainly interacts through its phosphate groups. In contrast, the AMP and ADP ribose and nitrogenous bases are internalized more often than ATP. Based on NMR-derived docking and molecular dynamics simulation, we propose an uptake mechanism in which the first step is the phosphate-binding process, and the second step is the flipping of the ribose and nitrogenous base that can change its orientation from a solvent-exposed to a buried state. The data obtained by NMR and molecular dynamics simulation provide insights into substrate uptake mechanisms and a basis for future drug design.

## 2. Materials and Methods

### 2.1. Protein Expression and Purification

The MtRpiB gene (locus Rv2465c) was cloned into a vector pCRT7/NT-TOPO, as previously described [11], and kindly provided by Annette Roos. The plasmid was transformed into *Escherichia coli* strain BL21(DE3), and freshly transformed colonies were grown overnight in 25 mL of Luria–Bertani (LB) medium containing 200 µg/mL of ampicillin until they reached an optical density (OD600) of 0.8 at 37 °C and stirring at 200 rpm. Then, 12.5 mL were transferred to a 2 L Erlenmeyer filled with 500 mL of M9 minimal medium ($Na_2HPO_4 \cdot 12H_2O$ 17 g/L, $KH_2PO_4$ 3 g/L, NaCl 0.5 g/L, $MgSO_4$ 0.24 g/L, glucose 3 g/L, $CaCl_2 \cdot 2H_2O$ 14.7 g/L, thiamine 1 g/L) containing 200 µg/mL of ampicillin and supplemented with 1 g/L 15NH4Cl. The cells were grown until they reached OD600 of 0.6, and the MtRpiB expression was induced by adding 1 mM of isopropyl-β-d-thiogalactopyranoside (IPTG) at 18 °C overnight. The cells were harvested by centrifugation ($10,000 \times g$ for 15 min at 4 °C) and stored at −20 °C.

For purification, the cells were resuspended in lysis buffer [50 mM Tris–HCl pH 8.0, 300 mM NaCl, 10 mM imidazole, 20% (*v/v*) glycerol, 1 mM phenylmethanesulfonyl fluoride (PMSF)] and disrupted in an ice-bath by sonication (225 W, 30 s pulses with 30 s intervals between pulses, 30 times). The lysate was clarified by centrifugation at $7000 \times g$ for 30 min at 4 °C, followed by supernatant filtration in a 0.22 µM membrane. This sample was purified with its injection in a HisTrap HP 5 mL column (GE Healthcare) that was equilibrated with buffer A (50 mM Tris–HCl pH 8.0, 500 mM NaCl, 10 mM imidazole, 10% (*v/v*) glycerol), and MtRpiB eluted with an 180 mL linear gradient of 10–500 mM imidazole. The MtRpiB fraction was concentrated to approximately 4 mL using a centrifugal ultrafiltration device (molecular weight cut-off 10 kDa, Merck-Millipore) and loaded into a Superdex 75 16/60 column (GE Healthcare) and equilibrated with NMR buffer (20 mM Tris–HCl pH 7.0, 100 mM NaCl, 5 mM DTT, 5 mM sodium azide, 5 mM EDTA and 2 mM PMSF). The eluted protein was concentrated, and purity was evaluated by SDS-PAGE, followed by an enzymatic assay by NMR to confirm that the enzyme was active (Figure S1).

### 2.2. Nuclear Magnetic Resonance

The binding assays via chemical shift perturbation were performed on a Bruker Avance III 800 MHz spectrometer, equipped with a 5 mm TXI triple-resonance probe. The protein resonances were previously assigned [15] and deposited in BioMagResBank (https://www.bmrb.wisc.edu/) (accessed on 5 January 2023) with Accession Number 50025. The temperature was the same as the assignment experiments, 303 K, ensuring the 2D [$^1$H, $^{15}$N] TROSY spectra. We ran gradient selection [$^1$H, $^{15}$N] TROSY spectra using echo–anti-echo TPPI for frequency discrimination in the indirect dimension. The spectral width was 16.0176 ppm for the $^1$H dimension and 34 ppm for the $^{15}$N indirect dimension. The spectra were acquired with 24 scans and 70 complex points in the indirect dimension. For the data analysis, the spectra were superimposed, the signal transferred comparatively, and the chemical shift perturbations (CSPs) measured by the program CCPNMR Analysis v.2.4.2 through the differences in the observed resonances of the atoms of 1H and 15N in the free and bound states: $CSP = \sqrt{0.5 \left[ (\Delta\delta_{1H})^2 + (0.15\Delta\delta_{15N}))^2 \right]}$. The tests were performed with ribose 5-phosphate, ribose, inorganic phosphate, adenosine monophosphate (AMP), adenosine diphosphate (ADP), and adenosine triphosphate (ATP). In the experiments, the protein concentration was 150 µM, ribose 5-phosphate 150 µM, AMP 200 µM, ADP 200 µM, ATP µM, and Pi 500 µM. The CSP caused by the ligand interaction was considered significant when the chemical shift difference ($\Delta\delta$) was greater than the $\Delta\delta$ average for all residues plus a standard deviation ($\Delta\delta > \Delta\delta_{av} + SD$).

In saturation transfer difference (STD) experiments, the protein concentration was 10 µM, and the concentration of the binding compound was 2 mM. In addition, the tests showed that the saturation transfer was more efficient at higher concentrations of D2O; therefore, acquisitions were performed at 66% D2O [16]. The AMP amplification factors

(ASTDs) were calculated according to Viegas et al. [17]. STD experiments were run at 18.8 T (800.4 MHz) using excitation sculpting for water suppression, the spectral width of 16.0176 ppm, 2 s of relaxation delay, 40 cycles of 8 scans, and a saturation field strength of 24.3 Hz (50 ms Gaussian-shaped pulse). STD experiments were performed using saturation times of 0.5, 1, 2, and 3 s and the saturation frequencies off-resonance of $-6000$ Hz ($-7.5$ ppm) and on-resonance of 217 Hz (0.27 ppm), 520 (0.65 ppm), and 5804 Hz (7.25 ppm). The difference spectra (ISTD) were obtained by subtracting the on- and off-resonance spectra. The following formula is applied to calculate the amplification factor in each of the protons (Equation (1)):

$$A_{STD} = \frac{I_{STD}}{I_0} \times \frac{[L]_T}{[P]} \tag{1}$$

where $A_{STD}$ is the amplification factor, $I_{STD}$ is the intensity difference, $I_0$ is the off-resonance pulse intensity, $[L]_T$ is the total ligand concentration, and $[P]$ is the protein concentration. The amplification factors were calculated qualitatively to the proximity of the ligand hydrogens with the protein. The value of 100% was stipulated for hydrogen with the most significant amplification factor, and the values for other hydrogens was calculated.

### 2.3. Site-Directed Docking

The HADDOCK (version 2.2) server [18] was used to construct a structural model of the complex of the *Mycobacterium tuberculosis* ribose 5-phosphate isomerase (MtRpiB) with adenosine monophosphate (AMP). The protein structure coordinates used as input values were obtained from the Protein Data Bank (PDB) under the access code 2VVP [13]. The chemical shift cross-peaks in the 2D 1H–15N HSQC spectra were defined as ambiguous interaction restraints at the AMP-binding interface. Residues with significant chemical shift perturbations ($\Delta\delta > \Delta\delta av + SD$) were classified as active from the solvent-accessible surface area calculations, while those neighbors of active residues were automatically defined as passives. The histidine protonation states were set according to the PROPKA results [19], considering a pH of 7.0. The scaling of intermolecular interactions for rigid body energy minimization was set to 0.01. The weights for the Van der Waals interaction of the rigid-body docking set and the electrostatic interaction of the water refinement stage were set to 0 and 0.1, respectively. The initial temperature for the second and third torsion angle dynamics (TADs) cooling steps with flexible sidechains at the interface and fully flexible interfaces were 500 and 300 K, respectively. The molecular dynamics (MD) steps for the rigid-body high-temperature TADs and during the first rigid-body cooling stage were set to 0. In total, 2000 complex structures of rigid-body docking results were executed using the standard HADDOCK protocol with an optimized potential for liquid simulation (OPLSX) parameters [20]. The final 200 lowest-energy structures were obtained for subsequent explicit solvent (water) refinement and semi-flexible simulated annealing refinement to optimize sidechain constraints. The structural conformation of the constructed model was displayed using PyMol software [21].

### 2.4. Molecular Dynamics Simulation

The molecular dynamics (MD) calculations were performed with the GROMACS (version 5.0.7) [22]. The molecular system was modeled with the CHARMM36 force field [23] and TIP3P water model [24]. The MtRpiB/AMP complex structure obtained from molecular docking calculations was used in the MD simulations. The complex structure with ADP was generated from the protein/AMP simulation frame at 100 ns, adding a phosphate group to the molecular structure of AMP. The structures of the protein/ligand complexes were placed at the center of an 82 Å cubic box solvated by a solution of 100 mM NaCl in water, and the protonation state of ionizable residues was set according to the PROPKA results [19], considering a pH of 7.0. Periodic boundary conditions were used, and all simulations were performed in an NPT ensemble, keeping the system at 298 K (25 °C) and 1.0 bar using a modified Berendsen thermostat ($\tau T = 0.1$ ps) and Parrinello–Rahman barostat ($\tau T = 2.0$ ps and compressibility = $4.5 \times 10^{-5}$ bar$^{-1}$). A cut-off of 14 Å for both

Lennard-Jones and Coulomb potentials was used. The long-range electrostatic interactions were calculated using the particle mesh Ewald (PME) algorithm. In every MD simulation, a time step of 2.0 fs was utilized, and all covalent bonds involving hydrogen atoms were constrained to their equilibrium distance. A conjugate gradient minimization algorithm was utilized to relax the superposition of atoms generated in the box construction process. Energy minimizations were conducted with the steepest descent integrator and conjugate gradient algorithm, using 1000 kJ·mol$^{-1}$·nm$^{-1}$ as the maximum force criterion. Finally, a 1.0 μs MD simulation was performed for the data acquisition. Following dynamics, the trajectories were concatenated and analyzed by different parameters, such as the number of hydrogen bonds (cut-off distance = 3.5 Å and maximum angle = 30°), number of contacts (<0.6 nm), and root-mean-square deviation (RMSD) of all non-hydrogen atoms for the ligands (AMP and ADP) and backbone atoms for the protein (MtRpiB). The contributions of MtRpiB residues to the nucleotide-binding energy were calculated from the MD trajectories using the Molecular Mechanics Poisson–Boltzmann Surface Area (MM-PBSA) method implemented in the g_mmpbsa program, along with the MmPbSaDecomp.py script [25,26].

### 3. Results

#### 3.1. The Phosphate Group Is Essential to the Interaction

The importance of ribose 5-phosphate (R5P) in nucleic acids and cofactors synthesis makes ribose 5-phosphate isomerase catalytic activity essential for some organisms. To better comprehend the mechanisms involved in substrate uptake within the active site, we performed several binding assays to identify sugar interaction. The structural features of the active site are relevant in the design of inhibitors for MtRpiB (Figure 1). The active site has an internal cavity permeable to water, but it is challenging to be transposable by larger molecules. The cavity is covered by a tightly bound phosphate ion, complexed by H102A, H138A, R141A, H12B, and R113B. Residues Q94A, L98A, A42B, and M114B participate in the coverage of the internal cavity. The design of new leads needs to consider the accessibility of the internal cavity.

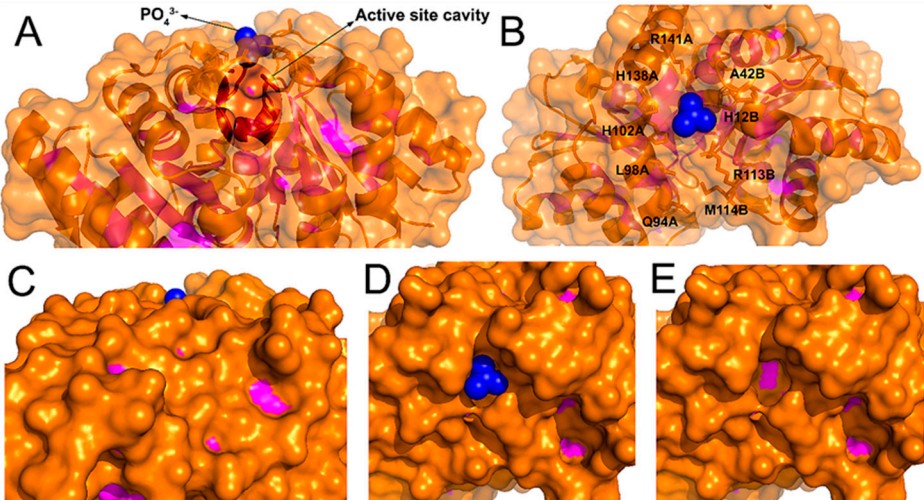

**Figure 1.** The active site of MtRpiB. (**A**) side view of the active site, highlighting the complexed inorganic phosphate and internal cavity. (**B**) Top view of the active site showing the surface residues. The active site is at the dimer interface, where residues come from subunits A and B. The residues in orange are exposed to the surface, while the magenta is buried. The semi-transparent surface enables the view of the secondary structure cartoon and side chains of the residues complexing the phosphate and covering the internal cavity. (**C**) Side view (**D**), top view, and (**E**) top view without the phosphate of the surface showing that the inorganic phosphate is exposed, almost entirely covering the internal cavity. In E, it is possible to observe the internal cavity, where the bottom residues are not exposed to the surface due to the concavity (magenta).

The interaction of R5P within the MtRpiB active site internal cavity is well-described. R5P can cross the internal cavity barrier and bind in the linear configuration of ribose to the interface between the two subunits. We performed a chemical shift perturbation (CSP) experiment using R5P as a ligand to confirm these data. The perturbation was analyzed based on previously assigned resonances [15], and the active site was confirmed (Figure 2A). We observed significant chemical shift perturbations for the following residues: Y7, D11, G14, E16, I33, A53, G65, Q76, A79, N80, L89, W91, A96, L107, H115, T116, V117, A122, R140, D143, I144, and A146. In addition, the resonances of residues G9, A10, H12, A13, Q19, D45, Y46, C50, A52, L68, G69, G70, G72, G74, E75, A99, E101, H102, G111, G112, R113, E119, I123, S133, H138, R141, I142, E147, and Y147 disappeared. Many of the observed CSPs are related to three influences. The first is the enzymatic activity of MtRpiB, which converts ribulose 5-phosphate to ribose 5-phosphate, and vice versa. This means that, at equilibrium, the protein has two ligands in the reaction mixture. The second, also correlated with catalytic activity, is the molecular dynamics of the catalytic process. This process can perturb the chemical shift of residues not directly involved with the binding process. Finally, it was not possible to conduct the experiments in a saturated condition. Thus, the protein is in equilibrium among free proteins, bound to R5P and ribulose 5-phosphate (Ru5P) during the data acquisition; this could explain the disappearance of many peaks in an intermediary exchange rate. However, the interaction was mapped at the active site or nearby areas. All residues interacting with the substrate in the crystallographic structure (PDB: 2VVP) disappeared or significantly changed their chemical shifts, except for N73. The second approach was to test sugar's ability to bind. The experiments containing pools with glucose, mannose, galactose, ribose, arabinose, xylose, fructose, glucosamine, fucose, and glucuronic acid showed no chemical shift perturbations (0.45 mM each, data not shown). To confirm this, we performed an assay containing a high concentration of ribose (4.5 mM). Nonetheless, the 30-fold excess did not cause significant perturbation in any amino acid (Figure 2B).

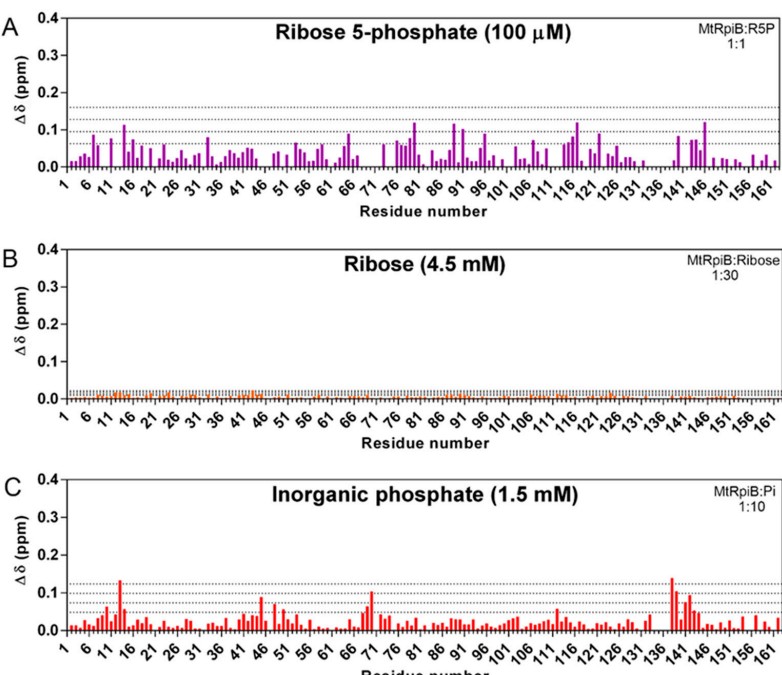

**Figure 2.** Chemical shift perturbation (CSP) of MtRpiB upon binding to the substrate. The ligand-induced CSP ($\Delta\delta$) was plotted for each amino acid measuring 2D [$^1$H, $^{15}$N] TROSY chemical shifts from free and bound proteins. The tested compounds were (**A**) ribose 5-phosphate, (**B**) ribose, and (**C**) inorganic phosphate (Pi). The dotted lines represent one, two, three, and four standard deviations above the averaged chemical shift changes of residues. We used 100 mM of MtRpiB in (**A**) and 150 mM in (**B**,**C**).

In contrast, the inorganic phosphate interacted with the protein (Figure 2C), evidencing its essential participation in the binding process. Thus, the mechanism of substrate uptake to the active site seems to have an initial stage, which depends on the phosphate interaction with residues located at the active-site entrance, followed by substrate internalization. The residues that showed significant CSP at 10× excess are A10, A13, G14, D45, A48, C50, G69, G70, G112, H138, Q139, R141, I142, and D143 (Figure 2C). Except for A48 and Q139, all other peaks also shifted in the presence of R5P. For a comparison with the CSP of R5P that we used, MtRpiB:R5P (1:1) (Figure 2A), we mapped the CSP at MtRpiB:Pi (1:3.3) and observed similar behavior, but with CSP more significantly reduced than the one observed for R5P (Figure S2).

*3.2. MtRpiB Interacts with AMP Derivatives*

Our first attempt, testing non-phosphorylated sugars, showed no interaction with MtRpiB; even the ribose showed no binding activity. In this way, nucleotide derivatives were tested to verify the binding activity. These compounds are pentoses with a nitrogenous base of at least one phosphate group, maintaining many similarities with R5P.

The interaction with adenosine monophosphate (AMP) showed exciting results (Figure 3B). The residues that suffered significant chemical shift perturbations were A10, D45, A48, L68, G69, G72, E75, Q76, A79, N80, N104, R113, H138, Q139, R141, I142, and D143, in addition to residues A13 and Y46 that disappeared. This result shows that AMP interacts with the target protein, even in the presence of a bulky nitrogenous base, suggesting that the internal cavity was reached and that it can be expanded. Additionally, the CSP profile is different when compared to inorganic phosphate. The most affected residue is E75, which is buried in the active site and is responsible for transferring protons between C1 and C2 of the substrates in the catalysis. This interaction suggests the internalization of the AMP molecule. However, the interaction with AMP is not associated with catalytic activity, and unlike a substrate, the ribose at the nucleotide is in the cyclic configuration, which could explain the smaller number of peaks that vanished with AMP when compared to the R5P interaction (Figure 3A). The adenosine diphosphate (ADP) interaction assay showed chemical shift perturbations similar to those observed for AMP. Residues A10, H24, D41, D43, A48, L68, G69, E75, Q76, N80, N104, G111, R113, R140, and R141 had their resonances modified (Figure 3C). The interaction with ATP modified the peaks D11, A13, G14, H24, D41, A42, D43, A48, H138, and Q139 (Figure 3D). The modifications caused by the nucleotides are similar to those caused by the interaction with inorganic phosphate (Figure 2C), except for the region between the residues L68-E75. This region, which contains the catalytic residues S71 and E75, seems to be less affected by the phosphate group isolated compared to the nucleotides. These modifications are similar to those caused by the interaction with inorganic phosphate (Figure 2C), except for D41, A42, and D43. These residues are in the loop region before the second β-strand, a region described in the literature as necessary for the interaction with other ligands. For a comparison with the CSP of R5P that we used, MtRpiB:R5P (1:1) (Figure 3A), we mapped the CSP at MtRpiB:nucleotide (1:1.5) and observed similar behavior, but with CSP (Figure S3). Figure S4 shows the [$^1$H, $^{15}$N] TROSY spectra at multiple nucleotides or R5P concentrations. Note that most CSPs are in the fast exchange regime, compatible with binding in the micro- to millimolar ranges.

The effect of the interaction of R5P and AMP in MtRpiB was highlighted in the protein structure (Figure 4). The R5P interaction causes an extensive change in the chemical shift, either by the shift or disappearance of peak resonances (Figure 4A). It can be related to the active-site open–close movements associated with the R5P interaction, the cis-enediol high-energy intermediate (HEI) formation, and the release of Ru5P. The effect of AMP interaction in the protein structure is subtler when compared to R5P, with the active-site region being the most affected (Figure 4B). Thus, the interaction with AMP may have a regulatory function, with inhibition most likely occurring through a mechanism of competition of both molecules for the active site.

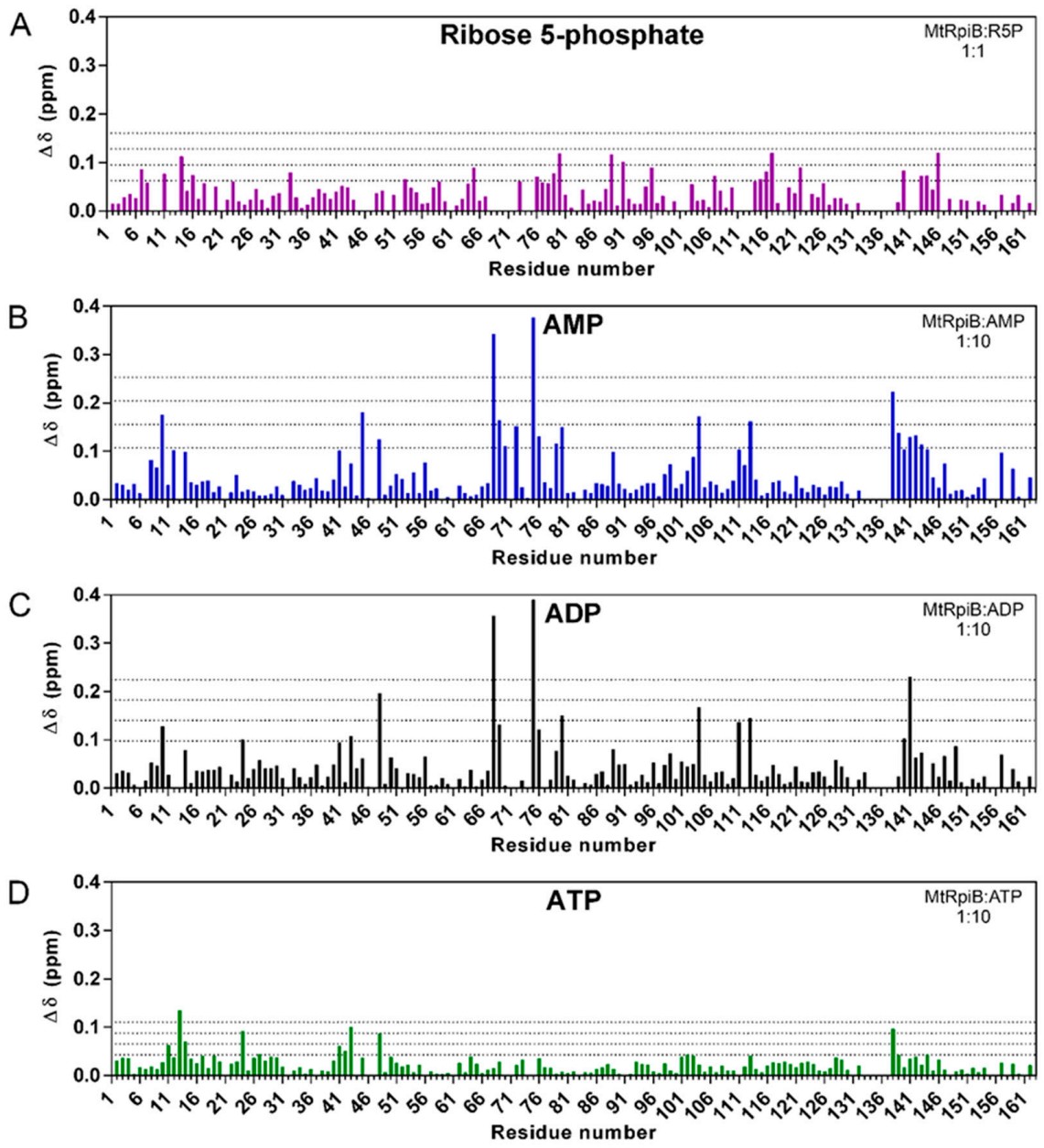

**Figure 3.** Chemical shift perturbation (CSP) of MtRpiB upon binding to R5P and AMP derivatives. The ligand-induced chemical shift perturbation (Dd) was plotted for each amino acid measuring 2D [$^1$H, $^{15}$N] TROSY chemical shifts from free and bound proteins. The tested compounds were (**A**) ribose 5-phosphate, (**B**) adenosine monophosphate (AMP), (**C**) adenosine diphosphate (ADP), and (**D**) adenosine triphosphate (ATP). The dotted lines represent one, two, three, and four standard deviations above the averaged chemical shift changes in residues.

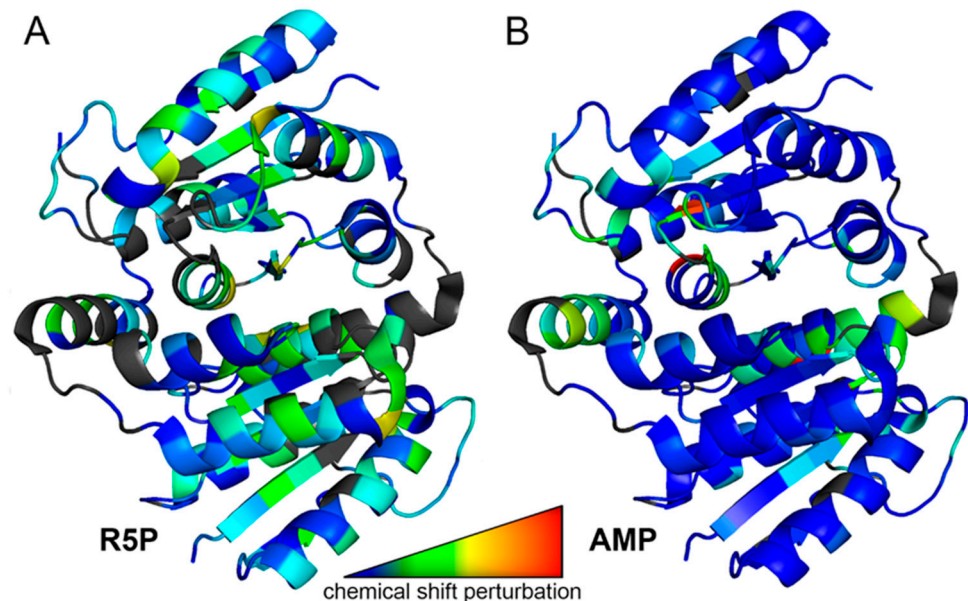

**Figure 4.** Chemical shift mapping of dimeric MtRpiB upon binding to (**A**) ribose 5-phosphate (R5P) and (**B**) adenosine monophosphate. The residues with significant chemical shifts (one standard deviation above the averaged chemical shift changes) were mapped on the protein structure. The perturbation was highlighted on the structure using a rainbow-colored spectrum with a dark-blue-to-red gradient. However, the residues that disappeared due to the interaction were colored gray (PDB code of MtRpiB: 2VVP).

### *3.3. The Interaction of Nucleotides with MtRpiB Has a Low Affinity*

The AMP:ATP ratio is an essential factor for energetic balance. The AMP and ADP increases are associated with a low-energy state inside the cells, and these molecules act as inhibitors that modulate the metabolic pathways. To investigate a possible biological role of AMP in MtRpiB inhibition, modulating the nucleotide and cofactor synthesis accordingly to energy levels, the dissociation constant (Kd) was determined using AMP, ADP, ATP, and inorganic phosphate compounds (Figure 5).

The AMP titration resulted in a significant perturbation of A10, H12, A13, A37, D41, D45, Y46, C50, L68, G69, G72, G74, N80, G111, G112, R113, H138, Q139, R141, and I142. The CSP experiment also identified these residues close to the active site (Figure 3B). The AMP global dissociation constant was 829 $\pm$ 138 $\mu$M, indicating a weak interaction (Figure 5A). For ADP, the titration significantly affected the residues A10, A13, G14, A37, D41, D43, A48, C50, L68, G69, and G72 (Figure 5B). Except for the N-terminus residues, these data are quite similar to those obtained from AMP. The global dissociation constant is 792 $\pm$ 115 $\mu$M, indicating the slightly higher affinity of ADP. In contrast, the binding affinity of MtRpiB to ATP was measured with a dissociation constant of 1.72 $\pm$ 0.833 mM, indicative of the decrease in the affinity. The residues that had a significant shift were A10, H12, A13, G14, A37, D41, D43, A48, C50, L68, G69, E75, G112, R113, H138, Q139, and R141 (Figure 5C). Finally, the inorganic phosphate (Pi) interaction affected the residues G9, A10, H12, A13, G14, D41, D45, C50, A53, L68, G69, G72, G112, H138, Q139, R141, I142, and D143, most of them similar to those previously identified with AMP. Meanwhile, the global dissociation constant for this compound is 2.78 $\pm$ 0.418 mM (Figure 5D). This value indicates that inorganic phosphate also shows a weak affinity, indicating that it is tight enough to avoid the entrance of non-phosphorylated sugars but is easily exchangeable; however, the addition of a ribose group and a nitrogenous base increased the affinity approximately three times for AMP and four times for ADP. Even with a lower affinity, the ATP dissociation constant is higher than Pi, demonstrating that these groups play an important role in the interaction. This result, along with the CSP data, indicates that the

nitrogenous base and ribose interact with the amino acid residues of the internal cavity of the active site.

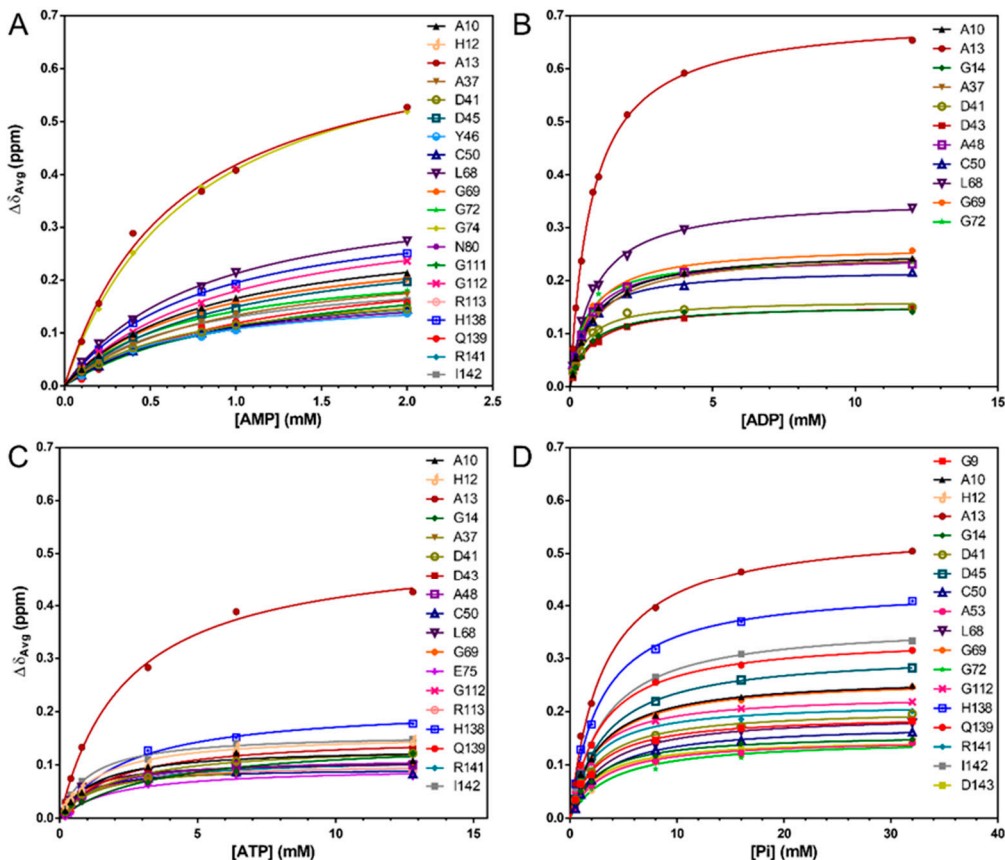

**Figure 5.** Determination of the dissociation constant of (**A**) adenosine monophosphate (AMP, Kd = 829 ± 138 mM), (**B**) adenosine diphosphate (ADP, Kd = 792 ± 115 mM), (**C**) adenosine triphosphate (ATP, Kd = 1.7 ± 0.8 mM), and (**D**) inorganic phosphate (Pi, Kd = 2.8 ± 0.4 mM) in complex with 150 μM MtRpiB. The chemical shift perturbation (Δδ) was plotted against increasing ligand concentration showing the residues with significant chemical shift perturbations (one standard deviation above the averaged chemical shift changes of residues). The HN peaks were traced in CCPNMR Analysis v.2.4.2 and fitted in Graph Pad Prism v.6. The [$^{1}$H, $^{15}$N] TROSY spectra are presented in Figure S3.

### 3.4. AMP Interaction with MtRpiB by Saturation Transfer Difference (STD)

The acquired STD spectra confirmed the interaction between MtRpiB and AMP (Figure 6). Additionally, based on the AMP assignment available in the Biological Magnetic Resonance Bank (BMRB) database, it was possible to identify the hydrogens in the compound that interact with the protein (Figure 6A,B). The STDs for all AMP hydrogens indicate that the compound is in close contact (≤5 Å) with the protein. Due to solvent suppression, the H2′ signal was lost by its proximity to water resonance. In addition, the STD amplification factor calculation determined the relative proximity of the region of the ligand to the protein (Figure 6C). The most efficient STD amplification was for H2 in the protein–AMP complex, followed by protons H1′, H4′, H3′, H8, and H10. These data suggest that the nitrogenous base can be entirely accommodated in the internal cavity of the active site. In addition, the STD data obtained from ADP and ATP also indicate that the nitrogenous base can be internalized (Figure S5).

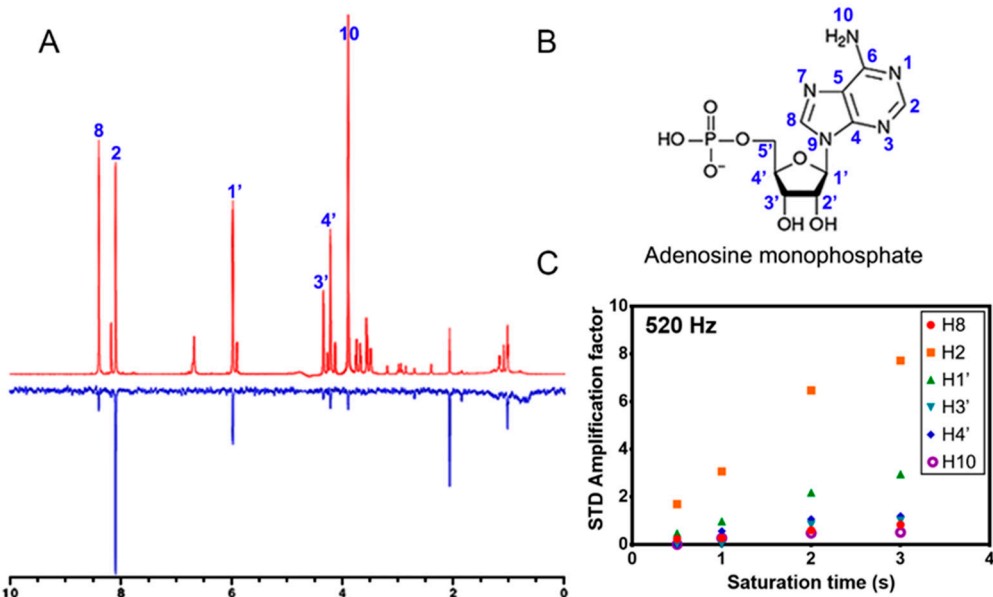

**Figure 6.** Saturation transfer difference (STD) NMR of MtRpiB interaction with adenosine monophosphate (AMP). (**A**) STD-NMR spectra (blue) and off-resonance spectra (red) were acquired with 10 μM MtRpiB titrated with 2 mM AMP. The AMP peak resonances were assigned and interacting hydrogens are annotated in blue. (**B**) AMP structure. (**C**) STD amplification factor calculated based on the intensity of the interacting protons. The saturation frequency used was 520 Hz (0.65 ppm) and the saturation times were 0.5, 1, 2, and 3 s. The data suggest the proximity of the ligand to the protein during the interaction (H2 > H1′ > H4′ > H3′ > H8 > H10).

### 3.5. Different Forces Drive the MtRpiB Interaction with R5P and AMP

The AMP was docked into the MtRpiB structure, and the result was compared to the X-ray structure of the complexed protein with R5P (Figure 7). In this structure, the interaction occurs with R5P in its opening chain conformation; the protein–ligand complex is stabilized after a nucleophilic attack that opens the furanose ring (Figure 7A,B). As previously mentioned, the enzyme interaction with the R5P phosphate group occurs through residues H12, R113, R137, and R141. In addition, the residues G70, S71, G74, H102, and N103 form hydrogen bonds in the 1,2-cis-enediolate high-energy intermediate, while residues D11, A13, Y46, and G69, and E75 participate in hydrophobic contacts and Van der Waals interactions. The residues that interact in the AMP are slightly different (Figure 7C,D). The roles of R113 and R137 in phosphate group coordination are maintained; however, interactions with H12 and R141 are not conserved. The MtRpiB interactions with those residues in R5P are replaced by two new hydrogen bonds with R113 and one with H102 in AMP. Evolutionarily, the enzyme interactions with Ru5P and R5P selected the formation of bonds with sugar in its open form or the furanose ring-opening as the first stage of the catalytic mechanism, followed by the stabilization of the complex through hydrogen bonds with an open form. Therefore, our results showing few hydrogen bonds between the enzyme and ribose group are consistent. The consequence is that the protein interaction with the nitrogenous base becomes extremely important for AMP maintenance inside the active site. The hydrogen bonds responsible for R5P stabilization in the protein–ligand complex are substituted by hydrophobic contacts and Van der Waals interactions with adenosine in AMP. These interactions occur through the residues D11, D45, G74, N103, R141, and H138, while the residues Y46, G69, G70, and E75 interact with the ribose group.

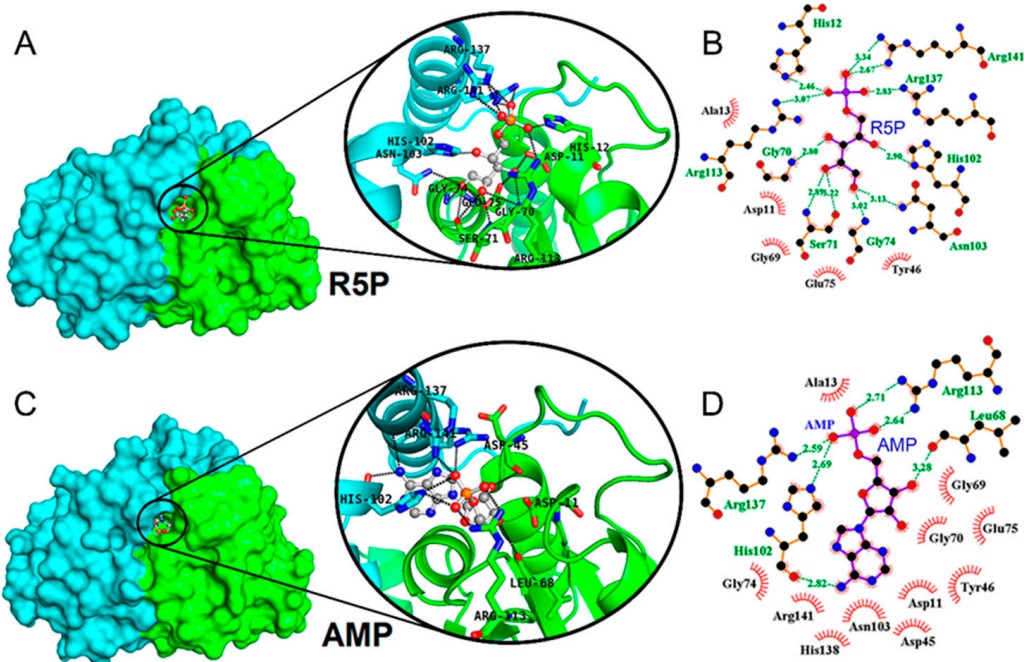

**Figure 7.** MtRpiB interaction with ribose 5-phosphate (R5P) and adenosine monophosphate (AMP). (**A**,**B**) MtRpiB X-ray structure in complex with R5P (PDB:2VVP). (**C**,**D**) AMP docked in MtRpiB active site. The stick representation highlights the protein–ligand interactions. In the MtRpiB complex with (**B**) R5P and (**D**) AMP, the carbon atoms are displayed in black, the nitrogen in blue, and oxygen in red. The hydrogen bonds are represented by green, dashed lines, with the distance in green, while the hydrophobic contacts and Van der Waals interactions are red, with the amino acid residues written in black.

### 3.6. A Flipping Mechanism Is Necessary for Ligand Internalization

We ran molecular dynamics (MD) simulations for 1 μs to analyze the fluctuations in the protein–ligand interactions for the structural model of the molecular complex (Figure 8A). The RMSD values for the MtRpiB backbone atoms indicate that the protein structure is stable throughout the simulation. However, when RMSD changes for the AMP variations are followed, the data indicate that this molecule undergoes significant structural fluctuations during the simulation time. These fluctuations are attributed to the internalization and exposition of the adenine and ribose in the enzyme active site several times during the simulation. The initial exposition of adenine to the solvent is identified by an increase in RMSD and a decrease in the number of hydrogen bonds and molecular contacts less than 0.6 nm (Figure 8A). These results indicate that the AMP phosphate group remains bound to the protein throughout the simulation, although the nitrogenous base can experience different conformations.

Based on the experimental data for ADP binding to MtRpiB, molecular dynamics simulations were performed in the presence of this compound. For this, another phosphate group was added to the AMP extremity already docked in MtRpiB, and after a step of energy minimization, this structure was used as the first frame of the molecular dynamics simulation (Figure 8C,D). This initial docking shows the ADP phosphate groups interacting with R137 and R141 of the protein. In addition, hydrophobic contacts and Van der Waals interactions are still responsible for stabilizing the interaction with ribose and adenine groups. At the end of the simulation, the last frame shows that the nitrogenous base leaves the active site and is solvent-exposed (Figure 8C,E). Although adenine has left the active site, the ADP molecule is still bound to the protein through hydrogen bonds with H102, H137, H138, and R141. The RMSD changes for the ADP variations throughout the simulation indicate that nitrogen-base exposition occurs in the middle of the simulation, at approximately 500 ns (Figure 8B). Once most hydrogen bonds occur with the phosphate

groups, they slightly vary over time. However, the number of contacts lower than 0.6 nm substantially decreases after base flipping outside the active site and remains low, up to 1 μs.

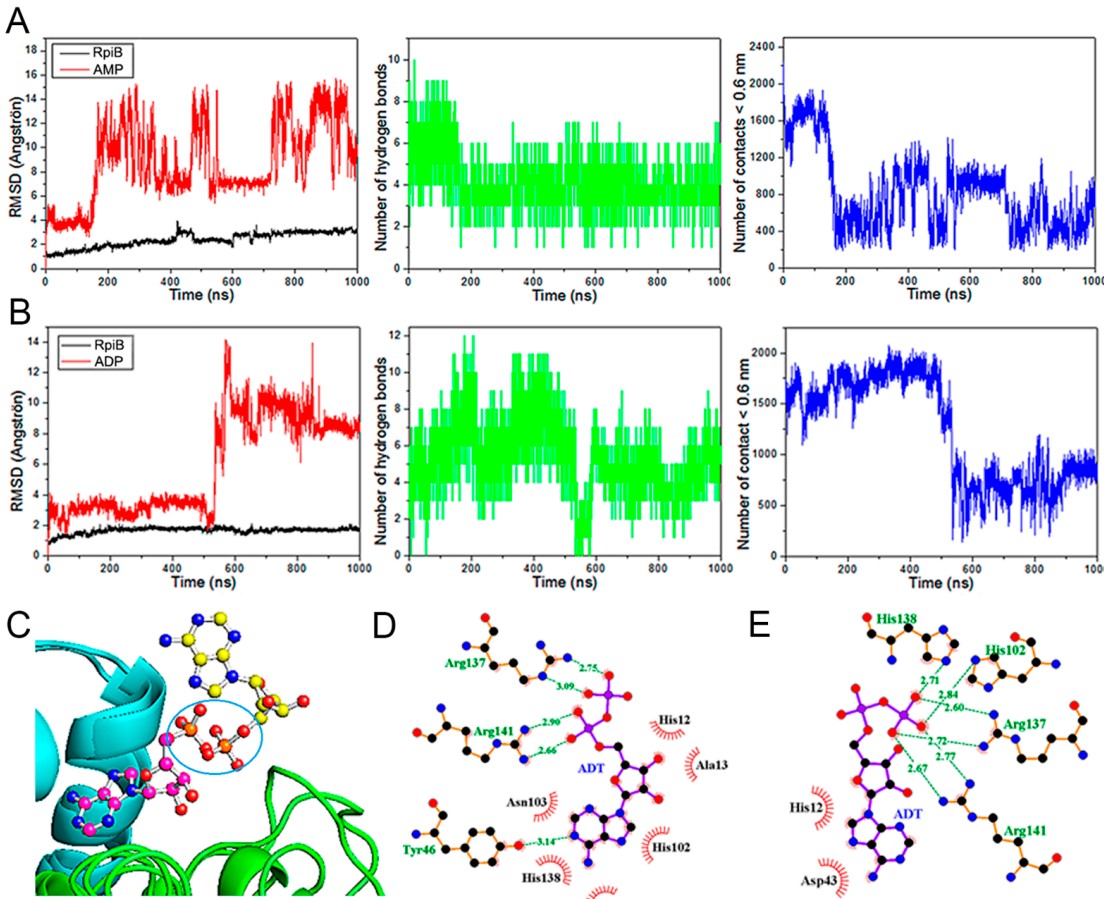

**Figure 8.** Molecular dynamics (MD) simulations of the structural models for the molecular complex of MtRpiB with adenosine monophosphate (AMP) and adenosine diphosphate (ADP). (**A**) MD simulations for MtRpiB–AMP complex. (**B**) MD simulations for MtRpiB–ADP complex. The interactions were analyzed following the RMSDs (red), the number of hydrogen bonds (green), and the number of contacts lower than 0.6 nm along 1 μs MD simulations. (**C**) Superposition of two models of the MD simulation of the MtRpiB–ADP complex at the beginning (inside the cavity) and end (outside de cavity) of the simulation, in which the phosphates of both conformers are superposed and highlighted by the blue circle. The phosphate atoms are displayed in orange, the nitrogen in blue, and oxygen in red. For better visualization, the carbon atoms are colored in magenta for initial conformation and nitrogen base inside the active site, or yellow for final conformation and nitrogen base exposed to the solvent. The ADP initial (**D**) and final (**E**) conformations are also displayed in a protein–ligand interaction layout. The color pattern of atoms and interactions are the same as Figure 6.

## 4. Discussion

We aimed to understand the necessary steps for the substrate and inhibitors' entrance at the active site of MtRpiB and to describe new inhibitors and their properties at the active site. The first attempt, using non-phosphorylated sugars, seemed promising since the substrates of this enzyme were phosphorylated sugars. Nevertheless, the results show no interaction with sugars, including excess ribose (Figure 2B). On the other hand, all the nucleotide assays resulted in some significant chemical shift perturbation (Figure 3B–D). The ATP perturbations appear to mainly result from the protein interaction with the ligand's phosphate group since the profile is more similar to that observed to bind inorganic phosphate (Figures 1C and 2C). The results for AMP and ADP seem to be more compatible

with those obtained for the interaction with R5P (Figure 2). However, the CSP values must be carefully compared with those obtained for the substrate since it was impossible to experiment with the same concentration of R5P. In addition, the residues that showed a more significant disturbance with the addition of R5P disappeared with the interaction, indicating an intermediate exchange regime. In any case, the analysis of the most perturbed residues indicates that AMP and ADP can be internalized in the active site. These data are reinforced by the result obtained by STD-NMR for the interaction with AMP (Figure 5), displaying an STD signal for all AMP protons, emphasizing the H2 of the nitrogenous base, indicating that this region is the one closest to MtRpiB during the interaction. The same STD pattern was observed for ADP and ATP internalizations (Figure S5).

The determination of the amplification factor obtained by STD data was used to map the ligand epitopes that were closest to the protein during the interaction, as described in the literature for several proteins, as in the case of O-acetylserine sulfhydrylase and the nucleoprotein of the Human Respiratory Syncytial Virus (hRSV), where this information was also used to perform directed docking [27,28]. Our data suggest that the phosphate group of a ligand or substrate is the first region to interact with the protein. This interaction proved essential for the ligand's uptake into the active-site cavity. The sugar interaction experiments showed that none of the ten non-phosphorylated sugars could bind to the protein. Even ribose, which is the carbon backbone of the substrate, did not interact with the protein when non-phosphorylated. We hypothesized that there was an energy barrier for the internalization of the unphosphorylated part of the substrate that could not be overcome without phosphorylation.

To better understand the characteristics of the interaction of the enzyme with AMP, molecular dynamic simulations were conducted for the structural model of the MtRpiB–AMP complex determined by a docking calculation, which was based on NMR data (CSP and STD). As the opening of the furanose ring is the first step in R5P catalysis and rapidly occurs at the beginning of the catalysis process [13], there is no experimental data in the literature for the protein–R5P complex in a closed form. However, an analysis of the crystallographic structure of the MtRpiB–R5P complex in an open form shows that most of the interactions that stabilize the complex are hydrogen bonds (Figure 7B). The NMR-derived docking of the protein–AMP complex showed that its stabilization was predominantly by hydrophobic contacts and Van der Waals interactions with the nitrogenous base. The few hydrogen bonds were mainly maintained to keep and stabilize the phosphate group bound to the protein (Figure 7D). Adding a new phosphate group to mimic the interaction with ADP indicates that hydrophobic contacts and Van der Waals interactions with the nitrogen base are still an important contribution to interactions and that hydrogen bonds occur with one or both phosphate groups (Figure 8D). Remarkably, our results suggest that the active-site cavity in the dimerization interface can expand and accommodate the nitrogen base. However, the chemical shift perturbation profile of ATP (Figure 3D) and its dissociation constant, when compared to those measured by the other nucleotides and inorganic phosphate, indicate the preference for phosphate group binding upon ribose and the nitrogenous base. Possibly, the three phosphates of ATP can be a barrier to the nitrogenous base and ribose internalization in the active site. We propose that a transition to a more internal phosphate is necessary before these groups can flip into the cavity.

The results of the MD simulation show that the interactions of AMP and ADP have peculiarities. These differences may explain the results of the CSP experiments with AMP, ADP, and ATP (Figure 3B–D). The slighter higher affinity of ADP concerning AMP can result from longer nitrogenous base permanence within the active site, probably due to the difference in the possibilities of the interaction with ADP phosphate groups. The interaction with two phosphate groups decreased the chance of the ribose and base leaving the internal cavity of the active site. The cavity occupation was more evident when we compared the total time of the molecular dynamics simulation since AMP entry, and exit in the active site occurred several times during the 1 μs simulation (Figure 8A).

In comparison, the simulation data of the ADP interaction show that the nitrogenous base does not have the same mobility in entering and leaving the active site (Figure 8B). The simulation was limited to two main steps. In the first stage, the base remained buried in the site, while the interaction with proximal and distal phosphate groups occurred through hydrogen bonds with residues R137 and R141. Eventually, the dynamics of the ADP favored the interaction only with the distal phosphate group through H102, R137, H138, and R141 residues. This change increased the loop-dynamics region closest to the active site, facilitating nitrogenous base removal from the active site and its exposure to the solvent. After the adenine left, it was not verified that it returned to the active site during the remaining 500 ns of the simulation. Although the nitrogenous base compounds were exposed and occluded from the solvent during the simulation time, the phosphate interactions were maintained in all experiments, keeping the ligands anchored to the protein (Video S1).

The interaction with ligands was essential to enrich our understanding of the catalytic mechanism of ribose 5-phosphate isomerase. The inhibitors 4-phospho-D-erythronohydroxamic acid (4PEH) and 4-phospho-D-erythronate (4PEA) were synthesized based on the structure of the intermediate cis-enediol (ato) [10] and provided details about the catalysis. From the results of the interaction of MtRpiB with AMP and inorganic phosphate, it is possible to understand the mechanism of substrate capture. Unlike R5P, the linear configuration of the ribose ring of the AMP was not possible. Therefore, we obtained information about its entry when creating the reverse path of the molecule's exit from the active site. We identified the need for an interaction with the phosphate group as essential for the beginning of the capture process. The phosphate group interacted with the arginine and histidine residues at the cavity entrance via hydrogen bonds, and ribose could rotate into the internal cavity of the active site. Our data suggest the inexistence of the proper open–close conformational equilibrium of the enzyme-active site. Probably, there is no open conformation, and the active-site dynamics provide a series of interactions that enable the flipping of AMP (and possibly the substrate) to reach the internal cavity without a fully open state. The internal cavity is a protected environment that stabilizes the intermediate state: the binding of the linear configuration of the furanose ring. To account for the phosphate contribution of the interaction of MtRpiB, we computed the theoretical Gibbs free energy of binding ($\Delta G_b$) using the Poisson–Boltzmann Surface Area (PBSA) (Figure S6). It enabled the discrimination of the residues that contribute to $\Delta G_b$. The behavior was similar for AMP and ADP and the most significant contributions were obtained from the charged residues distributed throughout the protein, but were more pronounced for those directly participating in the binding to phosphate, as illustrated in Figure S6C. There was also an asymmetric contribution from each dimer subunit. While chain A's contribution was unfavorable, chain B contributed favorably. The unfavorable contributions of D41, D43, D44, and D45 of chain A, which contributes to the asymmetry, were remarkable. The observed theoretical behavior was corroborated by the measured binding dissociation constants, for which we observed a $K_d$ of 829 ± 132 μM for AMP, 792 ± 115 μM for ADP, 1.72 ± 0.8 mM for ATP, and 2.72 ± 0.4 mM for inorganic phosphate (Figure 5), resulting in the following values of $\Delta G$: 14.6 kJ/mol for inorganic phosphate and 15.8 for ATP, while AMP and ADP were 29.0 and 29.1 kJ/mol, respectively. Therefore, the binding to phosphate alone contributes to approximately 50% of the energy necessary for binding. The conformational changes that drive nucleotide flipping arose from the asymmetry of the active site located at the dimer interface, resulting in asymmetric dynamics that enable the interaction of the nitrogenous base with residues at the internal cavity.

From a metabolic point of view, the interaction with AMP and ADP could be a mechanism for inhibiting the enzymatic activity of RpiB by competition. These compounds would be competitive regulators that signal a low-energy state; thus, an increase in AMP concentration would accelerate the metabolism in pathways that restore chemical energy, such as glycolysis and the citric acid cycle [29]. The literature reports that some proteins act as sensors of chemical energy levels and are responsible for restoring metabolic balance. An

example is AMP-activated protein kinase (AMPK), which, when interacting with AMP and ADP, stops being dephosphorylated and activates a catabolic pathway that results in ATP production [29,30]. In the case of RpiB, an enzyme from the beginning of the non-oxidative branch of the pentose phosphate pathway (PPP) that activates a biosynthetic pathway to produce nucleic acids, the AMP could signalize a low-energy state and trigger another inhibition mechanism of PPP. To verify this hypothesis, we performed enzyme inhibition assays. However, the low-molar-extinction coefficient of ribulose 5-phosphate and the high Michaelis constant (Km) of ribose 5-phosphate isomerase of Mycobacterium tuberculosis resulted in tests with considerable errors at the points where the substrate concentration was close to Km values between 0.5–4 mM, which are those described in the literature [31]. With the uncertainty of inhibition assays, we measured the dissociation constants (Kds) to obtain information on the affinity of MtRpiB to AMP and ADP. The Kd of AMP was estimated by NMR-STD, 530 μM (data not shown), and by CSP, 829 μM (Figure 5A). Additionally, the Kd of ADP was measured by CSP as 792 μM. These results indicate that the affinities are low, making it unlikely to have an inhibitory function in a biological context. Despite this, studies of absolute metabolite quantification have shown that the maximum intracellular concentration of AMP and ADP in *E. coli* in carbon starvation conditions is about 845 μM, a value close to those obtained for dissociation constants. Therefore, further studies are necessary to evaluate the biological relevance of these and other nucleotide mono- and diphosphates.

Although the possible physiological role of *Mycobacterium tuberculosis* ribose 5-phosphate isomerase's interaction with AMP still has to be elucidated, the data from NMR-derived docking and MD simulations were essential to identify how the substrate uptake mechanism occurs. The CSP experiments with the substrate, ribose, inorganic phosphate, and nucleotides (Figures 2 and 3) showed that the interaction with phosphate is essential for the substrate binding to the protein, this being the first step in the substrate uptake mechanism. Following this step, the substrate/AMP rotates into the internal cavity of the active site through the simple bond that unites the phosphate group to the ribose. This second step maintains the hydrogen bonds between the phosphate oxygen and arginine and histidine residues at the active-site entrance. Computational simulations for AMP and ADP showed that the protein interactions with the nitrogenous base are primarily hydrophobic contacts and Van der Waals, with these compounds being anchored to the active site by their phosphate groups. The dynamic behavior of AMP and ADP binding suggests that their nitrogenous base and ribose groups can be occluded or exposed to the solvent. In summary, the interaction of the nucleotides has a shared contribution of the phosphate interaction, which contributed with ~50% (−14.6 kJ/mol), and the nitrogenous base and ribose were responsible for the other ~50%. However, in the interaction with the substrate ribose 5-phosphate, the hydrogen bonds were predominant after the first step of catalysis, in which H102 nucleophilic attack results in the ring-opening of furanose (Figure 9). This mechanism and the internal cavity expansion should be taken into account when developing inhibitors of RpiB. Considering this, we proposed using phosphate mimetics, such as phosphonates and others [32], to perform screening with multiple ligands.

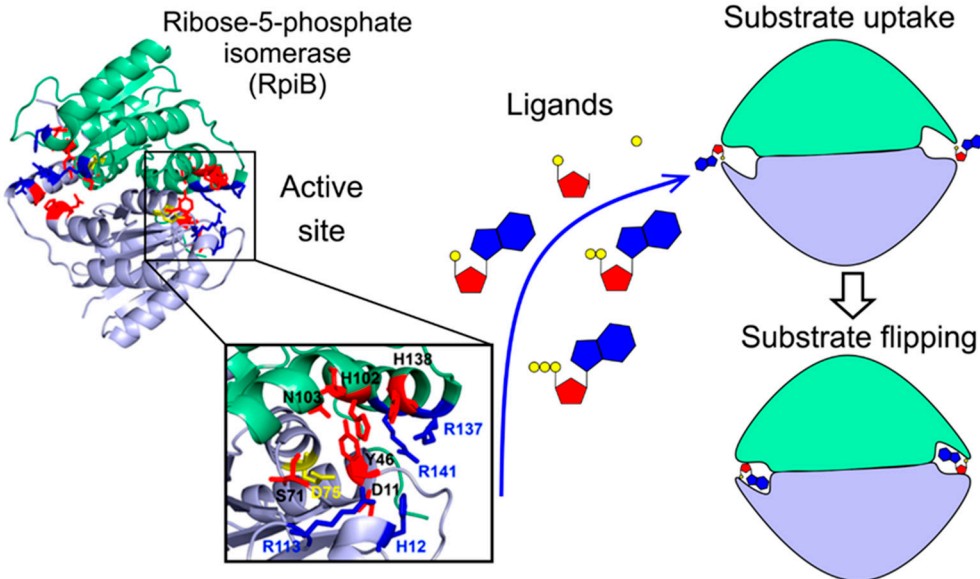

**Figure 9.** Scheme of the proposed substrate flipping mechanism. The essential amino acid residues at the active-site entrance interact with the phosphate group of the substrate. The substrate/ligand uptake is followed by the flipping movement in which the ligand is internalized. The compound size is variable, indicating that the active site displays some plasticity. Our data suggest that the enzyme can accommodate AMP and ADP more frequently than ATP, which indicates that the protein has to change into a more internal phosphate before the ribose and nitrogenous base are internalized.

**Supplementary Materials:** The following supporting information can be downloaded at: https://www.mdpi.com/article/10.3390/biophysica3010010/s1, Figure S1: Enzymatic activity of MtRpiB by 1D $^1$H NMR. (A) Ribose-5-phosphate (R5P) spectrum.; Figure S2: Chemical shift perturbation (CSP) of MtRpiB upon binding to the inorganic phosphate with 3.3× excess; Figure S3: Chemical shift perturbation (CSP) of MtRpiB upon binding to R5P and AMP derivatives; Figure S4: Raw data of Chemical Shift Perturbation (CSP) experiments of MtRpiB in the presence of R5P, AMP, ADP, ATP, and Pi; Figure S5: Saturation transfer difference (STD) NMR of MtRpiB interaction with adenosine monophosphate (AMP), adenosine diphosphate (ADP), and adenosine triphosphate (ATP); Video S1: Molecular dynamics (MD) simulation of the structural model for the molecular complex of MtRpiB with adenosine diphosphate (ADP); Figure S6: Energy contribution to Gibbs free energy change ($\Delta G_b$) for the binding of the nucleotides with MtRpiB.

**Author Contributions:** Conceptualization, L.B., Í.P.C., C.D.A. and F.C.L.A.; methodology, L.B., Í.P.C., B.M., J.R.M.P. and D.M.P.O.; validation, L.B., Í.P.C., C.D.A. and F.C.L.A.; formal analysis, L.B. and Í.P.C.; investigation, L.B.; resources, C.D.A. and F.C.L.A.; writing—original draft preparation, L.B.; writing—review and editing, L.B., Í.P.C., C.D.A. and F.C.L.A.; visualization, L.B., Í.P.C., C.D.A. and F.C.L.A.; supervision, C.D.A. and F.C.L.A.; project administration, C.D.A. and F.C.L.A.; funding acquisition, C.D.A. and F.C.L.A. All authors have read and agreed to the published version of the manuscript.

**Funding:** This research was funded by FAPERJ Grants 239229, 215141, 204432, awarded to FCLA, and 225356 awarded to CDAB and CNPq Grant 309564/2017-4, awarded to FCLA. We also thank INBEB-INCT for the funding. LB was funded by the CNPq scholarship and IPC was funded by FAPERJ (202280).

**Data Availability Statement:** MtRPIB chemical shift assignment information was deposited in BioMagResBank (https://www.bmrb.wisc.edu/ accessed on accessed on 5 January 2023) with Accession Number 50025.

**Acknowledgments:** We thank Annette Roos from Uppsala University for providing the plasmid containing the MtRpiB gene.

**Conflicts of Interest:** The authors declare no conflict of interest.

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
