# Peer review of "Insights into the Substrate Uptake Mechanism of Mycobacterium Tuberculosis Ribose 5-Phosphate Isomerase and Perspectives on Drug Development"

_biophysica, doi:10.3390/biophysica3010010_

Round 1

Reviewer 1 Report

The research paper presents the novel work and has potential interest to researchers in the related field. The authors have provided sufficient background information followed by a clear and concise methodology. The results are well presented with a sufficient discussion of the presented results. To sum up, the research presented in this manuscript is well-designed and has novelty. I, therefore, recommend the manuscripts for publication.

Author Response

We thank the reviewer for the positive evaluation

Reviewer 2 Report

In this study, the authors show by MD simulation that the MtRpiB/Nucleotide (AMP/ADP/ATP) interaction involves a flipping mechanism. The results are supported by NMR experiments such as CSP and STD. Furthermore, they show that the phosphate moiety plays a key role in the interaction mechanism. Moreover, the MtRpiB/Nucleotide interaction suggests that AMP/ADP/ATP physiological ratio could contribute to the MtRpiB catalytic activity regulation.

The research is original and provides new insights to the interaction mechanism of MtRpiB. However, some points deserve major revisions.

- Line 124: the authors have performed an enzymatic assay by NMR to evaluate the MtRpiB activity. If it is a standard test, the authors must put the reference. However, if it is a new assay, developed by the authors, they must describe the experimental conditions and show the results.

- Nuclear magnetic resonance section (line 127): please indicate the NMR registration parameters, number of scans, number of increments, spectral width... For STD experiments, provide on/off-resonance frequencies in ppm, saturation power or saturation field strength, water and protein signals suppression parameters.

- Line 136: the authors should indicate the 15N chemical shift normalization factor employed to calculate the 1H-15N CSP.

- Line 144: the authors mention that saturation transfer is more efficient at higher concentrations of D2O, is there any reference to this phenomenon?

- In figure 2 and 3, the authors show the CSP induced by the interaction with R5P (1/1 ratio), AMP (1/10 ratio), ADP (1/10 ratio), ATP (1/10 ratio) and inorganic phosphate (1/10 ratio). However, it is clear from Figure 5 that ligand saturation has not been reached for these concentrations. It would be better to analyze the CSP under saturation conditions, especially since the data is already acquired (Figure 5). Therefore, that should also improve the identification of interface residues used for the docking process.

I strongly recommend to provide the 2D TROSY spectra in the presence and absence of the ligands in the supporting data.

Regarding the interaction with R5P, the MtRpiB/R5P ratio is too low (1/1), the exchange between the R5P-bound state, Ru5P -bound state, and unbound state can cause additional line broadening, which could complicate the CSP analysis. The authors indicate Lines 246 and 463, that experiments at higher concentrations of R5P could not be performed. The authors should explain why? Experiments under R5P/Ru5P saturation conditions might be more suitable for CSP analysis. The R5P/Ru5P ratio at equilibrium should also be determined.

Line 240: “A large number of modifications are related to some factors.” This sentence needs to be rewritten.

Line 280: Authors indicate that A13 and Y46 signals disappear with the interaction, however, A13 and Y46 signals were used for the CSP titration (Figure 5). Is there an intermediate exchange at that concentration? please explain that point.

Line 286. ´´However, the interaction with AMP is not associated with catalytic activity, and different from a substrate, the ribose at the nucleotide is in the cyclic configuration explaining the smaller number of peaks that vanished with AMP when compared to R5P interaction (Figure 3A).´´ The authors explain why fewer signals disappear, however, this is a hypothesis, they should use conditional tense

Line 332. Please provide STD titration curves for KD determination.

Line 365-368: ´´In addition, the STD amplification factor calculation determined the relative proximity of the ligand protons to the protein (Figure 6C). Thus, the result indicates that H2 is the closest AMP proton in the protein-ligand complex, followed by protons H1', H4', H3', H8 and H10.´´ - Group epitope mapping by STD identifies segments of ligands in direct contact with the protein by ranking the STD effect, it does not provide individual hydrogens proximities. Please review Mayer and Meyer JACS -2001. Authors should carefully discuss this point.

Line 481 “Even ribose itself, which is the primary substrate component group and undergoes many conformations during the catalysis, does not interact with the protein when non-phosphorylated, even at high concentrations.” This sentence needs to be rewritten.

Line 557-564: “To verify this hypothesis, we performed enzyme inhibition assays. However, the low molar extinction coefficient of ribulose-5-phosphate and the high Michaelis constant (Km) of ribose-5-phosphate isomerase of Mycobacterium tuberculosis resulted in tests with much error at the points where the substrate concentration is close to the Km values between 0.5 - 4 mM, which are those described in the literature [28]. With the uncertainty of inhibition assays, we measure the dissociation constants (Kd) to obtain information on the affinity of MtRpiB to AMP and ADP.”

The authors may try R5P/Ru5P EXSY NMR enzymatic assays,

Reviewer 3 Report

In the current manuscript, the authors have delineated the mechanism of the binding of ribose -5’- phosphate to M. tuberculosis ribose-5’-phosphate isomerase, using NMR spectroscopy and molecular dynamics simulations. The results are well presented, however, there are some shortcomings that the authors need to address to make the manuscript suitable for publication.

1.     The major finding reported in the manuscript is the mechanism of substrate binding wherein, the authors have highlighted that phosphate binding precedes the flipping of the substrate at the active site. It is not clear what are the conformational changes in the protein that drives substrate flipping. Does phosphate binding provide enough energy for substrate flipping?

2.     In all panels of Figure 3, the authors have indicated the molar ratios between MtRpiB and ribose-5’ -phosphate/adenosine nucleotides used for the measurement of CSPs. In the discussion, the authors have alluded to the fact that the concentration of ribose- 5’- phosphate could not be increased to levels comparable to that of the adenosine nucleotides. In that case, the authors need to fix the nucleotides at a concentration that is the same as that of ribose-5’ -phosphate and check if the CSPs are similar to those estimated at high nucleotide concentrations. The same is true for Figure 2, wherein the authors should check CSPs at the same concentrations for all ligands.

3.     In Figure 4, the authors have mapped the residues undergoing chemical shift perturbations, onto the structure of MtRpiB and concluded that AMP competes with ribose- 5’-phosphate for binding to the active site. One of the approaches that the authors can take to validate competitive binding is to monitor chemical shift perturbations by varying ribose-5’-phosphate using fixed concentrations of the adenosine nucleotides. The drop in affinity of ribose-5’-phosphate with the increase in the concentration of adenosine nucleotides will indicate competition between the two ligands.

4.     In the NMR methods section the authors should elaborate on how they have estimated the global dissociation constants. Also, the estimation of the dissociation constant for ATP has a huge error. Hence, the conclusion, ATP has a weaker affinity than AMP or ADP (lines 336-338), is not correct. The authors should report the dissociation constants using techniques like ITC, microscale thermophoresis, biolayer interferometry, or surface plasmon resonance, in addition to the estimation by CSPs.

5.     In Figure 5, ADP and ATP have been varied up to concentrations higher than 10 mM. Usually, nucleotides tend to precipitate at such high concentrations. Do the NMR peaks corresponding to ADP and ATP, keep increasing in intensity up to 10 mM or more? This will inform on the solubility of the nucleotides at the concentration ranges used in the titrations.

6.     In the STD data (Figure 6A and Supplementary Figure 1), the difference spectra contain peaks with negative intensities. Usually, peaks with positive intensities are expected after subtracting the on-resonance and off-resonance spectra. The authors need to explain what resulted in negative peaks upon subtraction.

7.     The authors have used 3 different saturation times and on-resonance frequencies for the acquisition of STD-NMR spectra. However, there is no mention of the on-resonance frequency which was used for the data presented in the manuscript. The authors need to include this information in the methods.

8.     The citation of reference 28 is not appropriate. The authors have cited this reference for the high Km of M. tuberculosis ribose-5’-phosphate isomerase, for its substrate (lines 557-562). The cited review does not contain any information on isomerases.

Round 2

Reviewer 2 Report

The authors have responded satisfactorily to all the comments. The changes made have greatly improved the paper. The paper can be accepted in its current form.

Author Response

We thank the reviewer for the positive and constructive evaluation.

Reviewer 3 Report

1.     In the author’s response to question 2, similar concentrations of the ligands have been used for probing the CSPs upon ligand binding to ribose 5’-phosphate isomerase. However, in the manuscript, the authors have mentioned that they have used different molar ratios in the case of inorganic phosphate and ribose 5’ phosphate (lines 277-280). Are the molar ratios expressed as the ratio of quantities of protein and the ligand?

2.     In the response to question 3, the authors mentioned that they have provided the competition statement in line 346. But the sentence is not found in the manuscript. The authors need to clarify this omission or mention the correct location of the sentence in the manuscript.

Author Response

Reviewer 3:

  1. In the author’s response to question 2, similar concentrations of the ligands have been used for probing the CSPs upon ligand binding to ribose 5’-phosphate isomerase. However, in the manuscript, the authors have mentioned that they have used different molar ratios in the case of inorganic phosphate and ribose 5’ phosphate (lines 277-280). Are the molar ratios expressed as the ratio of quantities of protein and the ligand?

The molar ratios are expressed as the ratio MtRpiB:Ligand. We reported for inorganic phosphate the ratio 1:3.3 (Suppl. 2) and 1:10 (Figure 2C). For ribose 5´ phosphate, we have used the ratio 1:1 (Figure 2A).

To make it clearer, in the revised version, we added the following statement to Figure 2 legend: “We used 100 mM of MtRpiB in A and 150 mM in B and C.”

  1. In the response to question 3, the authors mentioned that they have provided the competition statement in line 346. But the sentence is not found in the manuscript. The authors need to clarify this omission or mention the correct location of the sentence in the manuscript.

The sentence is actually in line 335: “Thus, the interaction with AMP may have a regulatory function, with the inhibition occurring most likely through a mechanism of competition of both molecules for the active site.”